# CROME: Covariate-Balanced Causal Representation Learning for Composite Outcomes via Multi-Task Estimation in Electronic Health Records

## Abstract

Estimating treatment effects on composite outcomes is challenging, particularly in high-stake decision making domains such as healthcare where multiple related outcomes jointly inform clinical decisions. Existing approaches often simplify this problem by collapsing multiple component outcomes into a single target, overlooking the underlying structure, introducing modeling bias, and limiting interpretability. In this work, we propose **CROME** (**C**ausal **R**epresentation for Composite **O**utcome via **M**ulti-task **E**stimation), a framework that leverages representation learning, multi-task learning (MTL), and covariate-balancing techniques to predict component-level potential outcomes, which are then aggregated through a user-specified utility function. CROME jointly learns a shared representation across tasks along with outcome-specific prediction heads, enabling accurate and interpretable estimation of treatment effects on composite outcomes. Our theoretical results show that CROME achieves lower generalization error, under mild conditions, than MTL without shared representation and single-task baselines. Empirical results on synthetic and semi-synthetic datasets inspired by the Infant Health and Development Program (IHDP) and an electronic health records (EHR) dataset in oncology confirm the advantages of our approach over existing methods including enhanced accuracy and interpretability. Our framework provides a principled and flexible solution for causal inference in complex, multi-outcome clinical settings, with broad applicability across patient-reported and EHR-derived data.

## 1 Introduction

Patient-reported outcomes (PROs) (Deshpande et al., 2011; Rothman et al., 2007; Lohr & Zebrack, 2009; Johnston et al., 2019; Fayers & Machin, 2013) are health reports given directly by patients about how they feel and how a treatment affects their daily life. They play an important role in evaluating treatment efficacy by capturing patients' perspectives on symptoms, functioning, and quality of life. In clinical research domains such as oncology and chronic disease management, multiple PROs are often aggregated into a composite outcome (Cordoba et al., 2010; McKenna & Heaney, 2020; Freemantle et al., 2003; Gewandter et al., 2021; Wells et al., 2021) to provide a holistic measure of patient well-being. These composite outcomes are typically defined through a utility function (Luckett et al., 2021) that reflects clinical priorities or patient preferences and are increasingly used as primary endpoints in trials. However, estimating causal effects on composite outcomes remains challenging, particularly when component outcomes share complex latent structure or when the data are noisy, as in electronic health record (EHR) settings (Chauhan et al., 2025).

A central difficulty lies in the fact that composite outcomes obscure the underlying structure and heterogeneity of their component outcomes. Directly modeling the composite can lead to misspecified models and poor generalization, particularly when the function linking components is complex. Moreover, PROs and EHR-derived features often share latent structure across outcomes, making it inefficient to estimate each component separately. These challenges call for new methodological

advances that both respect the structural relationships among component outcomes and leverage shared information across prediction tasks.

To address these challenges, we propose **CROME**—a novel framework for Causal Representation Learning for Composite Outcomes via Multi-Task Estimation. Our key contributions are as follows:

- **New framework for composite outcome modeling**: We introduce a multi-task learning architecture with a shared encoder and task-specific heads as well as covariate-balancing regularization to predict individual potential outcomes. These predictions are composed using a *user-defined utility function*, allowing flexible and interpretable modeling of the treatment effect on composite outcome.

- **Theoretical guarantees for generalization error in treatment effect estimation**: We derive *generalization bounds* for the treatment effect estimation on composite outcome via CROME and prove that CROME yields *tighter error bounds* than both single-task and multi-task learning without shared representation baselines under mild conditions, particularly in settings with correlated outcomes.

- **Clinically meaningful interpretability**: By predicting each component outcome and explicitly modeling their aggregation, CROME enables *decomposition of the treatment effect on composite outcome* into component-level contributions, facilitating interpretation at both individual and cohort levels.

- **Extensive empirical validation**: We evaluate CROME on synthetic benchmarks, IHDP, and semi-synthetic EHR data, showing that it consistently outperforms several baselines in *accuracy, and interpretability* of treatment effect estimation.

## 2 RELATED WORKS

**Causal learning and treatment effect estimation.** A large body of research has focused on estimating CATE from observational data. Classical approaches include meta-learners (Künzel et al., 2019; Nie & Wager, 2021), doubly robust methods (Kennedy, 2023), tree-based techniques such as Causal Forests (Athey & Wager, 2019) and Bayesian Additive Regression Trees (BART) (Chipman et al., 2010), as well as neural network-based architectures including TARNet and CFRNet (Shalit et al., 2017). More recent innovations explore generative modeling and embedding-based frameworks to enhance treatment effect estimation (Makino et al., 2022; Rivera et al., 2023; Wu et al., 2024). In parallel, causal representation learning has emerged as a promising approach for mitigating bias by learning balanced or invariant representations of covariates, building upon the foundational potential outcomes framework (Rubin, 1974; Rosenbaum & Rubin, 1983). Notable advances include interventional and invariant representation learning approaches (Sun et al., 2024; Chauhan et al., 2025; Ahuja et al., 2023; Schölkopf et al., 2021), which aim to capture causal structure while improving predictive accuracy.

**Causal effect estimation and multi-task learning.** Multi-task learning (MTL) has been widely studied to improve generalization by leveraging shared information across related prediction tasks (Zhang & Yang, 2021; Maurer et al., 2016). In the context of causal inference, recent efforts (Alaa & Van Der Schaar, 2017; Aglietti et al., 2020; Jiang et al., 2023) have explored using MTL frameworks to enhance treatment effect estimation. By jointly modeling multiple related treatments, MTL can reduce sample complexity and exploit latent structure, thereby improving estimation efficiency and robustness.

Despite advances in causal representation and multi-task learning, their integration for estimating treatment effects on composite outcomes is underexplored. Composite outcomes—utilities over multiple endpoints—pose structural and clinical challenges. We propose a covariate-balancing aware, multi-task causal representation framework that shares representations across outcomes and aligns inference with the utility, improving accuracy and interpretability.

## 3 PRELIMINARIES AND NOTATIONS

To fix ideas, consider a binary treatment setting with covariates $x \in \mathcal{X}$, where $\mathcal{X}$ denotes the covariate space. The treatment assignment is represented by a binary variable $A \in \{0, 1\}$, where $A = 1$

denotes treatment and $A = 0$ denotes control. For each component outcome $k \in \{1, \ldots, K\}$, we denote by $Y_k(a) \in \mathbb{R}$ the potential outcome under treatment level $a$. The corresponding predicted potential outcome based on covariates $x$ is denoted $\hat{Y}_k^{(a)}(x)$. We also define the composite potential outcome under treatment $a$ as $U(a) = u(Y_1(a), \ldots, Y_K(a))$, where $u : \mathbb{R}^K \to \mathbb{R}$ is a user-specified utility function that maps component outcomes into a scalar. Its estimate is given by $\hat{U}^{(a)}(x) = u(\hat{Y}_1^{(a)}(x), \ldots, \hat{Y}_K^{(a)}(x))$. The CATE on the composite outcome is defined as $\tau(x) = \mathbb{E}[U(1) - U(0) \mid x]$, the average treatment effect (ATE) is defined as $\tau = \mathbb{E}[U(1) - U(0)]$, and their corresponding estimators are $\hat{\tau}(x) = \hat{U}^{(1)}(x) - \hat{U}^{(0)}(x)$ and $\hat{\tau} = \sum_{i=1}^n \hat{\tau}(x_i)/n$ where $n$ is the number of observations. We assume the utility function $u$ is $L_u$-Lipschitz with respect to the $\ell_1$ norm. A natural and widely used example of a user-specified utility function is the *Boolean composite function*: $U(a) = 1 - \prod_{k=1}^K (1 - Y_k(a))$, where $Y_k(a) \in \{0, 1\}$ is a binary indicator of whether component outcome $k$ occurs under treatment $a$. This utility captures whether *any* of the component events occurs under treatment $a$, and is equivalent to the logical "OR" operation over the individual outcomes. That is, $U(a) = 1$ if at least one $Y_k(a) = 1$, and $U(a) = 0$ only if all outcomes are zero. We show that the Boolean composite function is 1-Lipschitz in the Appendix A. Additionally, we also assume the strong ignobility (Rosenbaum & Rubin, 1983) and positivity assumptions.

**Modeling, Empirical Loss, and Balance Regularization.** Denote the observed data by $\{(x_i, a_i, y_{i1}, \ldots, y_{iK})\}_{i=1}^n$, where $x_i \in \mathcal{X}$ is the covariate vector, $a_i \in \{0, 1\}$ is the treatment, and $y_{ik} = Y_k(a_i)$ is the observed value of the $k$-th outcome component for unit $i$; assume $n \gg K$. Each (counterfactual) component outcome is predicted via $\hat{Y}_k^{(a)}(x) = f_k^{(a)}(\Phi(x))$, where $\Phi : \mathcal{X} \to \mathbb{R}^D$ is a shared representation (encoder) and $f_k^{(a)} : \mathbb{R}^D \to \mathbb{R}$ is a task-specific head. Let $\mathcal{I}_a = \{i : a_i = a\}$ and $n_a = |\mathcal{I}_a|$. With a loss $\ell(\cdot, \cdot)$ that is $L$-Lipschitz in $\ell_1$, the empirical loss for predicting $Y_k(a)$ is $\mathcal{E}_k^{(a)} := \frac{1}{n_a} \sum_{i \in \mathcal{I}_a} \ell\left(\hat{Y}_k^{(a)}(x_i), y_{ik}\right)$. The overall predictive loss across outcomes and samples is $\mathcal{L}\left(\Phi, \{f_k^{(0)}, f_k^{(1)}\}_{k=1}^K\right) = \frac{1}{n} \sum_{i=1}^n \sum_{k=1}^K \ell\left(\hat{Y}_k^{(a_i)}(x_i), y_{ik}\right)$. To mitigate distributional imbalance between treated and control groups in the latent space, we add a *balance regularizer* that penalizes divergence between the representations of treated and control units. Let $\mathcal{H}_a := \{\Phi(x_i) : i \in \mathcal{I}_a\}$ for $a \in \{0, 1\}$. Define $\mathcal{R}_{\text{bal}} := D(\mathcal{H}_1, \mathcal{H}_0)$, where $D$ can be chosen such as the maximum mean discrepancy (MMD):

$$\text{MMD}^2(\mathcal{H}_1, \mathcal{H}_0) = \left\| \frac{1}{n_1} \sum_{i \in \mathcal{I}_1} \varphi(\Phi(x_i)) - \frac{1}{n_0} \sum_{i \in \mathcal{I}_0} \varphi(\Phi(x_i)) \right\|_{\mathcal{H}}^2,$$

with feature map $\varphi$ (e.g. linear kernel or Gaussian RBF) into a Reproducing Kernel Hilbert Space. Model training is then proceeds by $\min_{\Phi, \{f_k^{(0)}, f_k^{(1)}\}_{k=1}^K} \mathcal{L}_{\text{total}}$, and $\mathcal{L}_{\text{total}}$ is the final training objective augments the predictive loss with this balance term:

$$\mathcal{L}_{\text{total}} = \mathcal{L}\left(\Phi, \{f_k^{(0)}, f_k^{(1)}\}_{k=1}^K\right) + \lambda \, \mathcal{R}_{\text{bal}} = \sum_{k=1}^K \left(\mathcal{E}_k^{(0)} + \mathcal{E}_k^{(1)}\right) + \lambda \, \mathcal{R}_{\text{bal}},$$

where $\lambda > 0$ trades off predictive accuracy and representation balance.

**Function Classes and Complexity.** We now introduce the function classes used in our generalization analysis. Let $\mathcal{F}_k$ denote the hypothesis class for independently modeling outcome $Y_k(a)$ without a shared representation. Let $\mathcal{H}_k$ denote the class for head functions $f_k^{(a)}$ applied after the shared encoder $\Phi$, which is drawn from class $\mathcal{G}$. For single-task composite models, the corresponding hypothesis class is denoted $\mathcal{F}_u$. To characterize model capacity, we use empirical Rademacher. The empirical Rademacher complexity of a function class $\mathcal{F}$ over a dataset $\{x_1, \ldots, x_n\}$ is given by:

$$\mathcal{R}_n(\mathcal{F}) := \mathbb{E}_\sigma \left[ \sup_{f \in \mathcal{F}} \frac{1}{n} \sum_{i=1}^n \sigma_i f(x_i) \right], \quad \sigma_i \sim \text{Uniform}\{-1, +1\}.$$

The complexity measure will be used in our generalization analysis to compare shared representation learning with alternative modeling approaches.

**Task-Averaged Risk and Complexity of Shared Representations.** To quantify the complexity of the shared representation, define a random set $\mathcal{H}(\bar{\mathbf{X}})$ as:

$$\mathcal{H}(\bar{\mathbf{X}}) = \{(\Phi_d(X_{ki})) : \Phi \in \mathcal{G}\},$$

where $X_{ki}$ is the covariate for unit $i$ and outcome $k$, and $\Phi_d$ is the $d$-th coordinate of the encoder output. The Gaussian average of this class is:

$$G(\mathcal{H}(\bar{\mathbf{X}})) := \mathbb{E}\left[\sup_{\Phi \in \mathcal{G}} \sum_{dki} \gamma_{dki} \Phi_d(X_{ki}) \,\middle|\, X_{ki}\right],$$

where $\gamma_{dki} \sim \mathcal{N}(0,1)$ are i.i.d. standard normal variables. For many function classes (e.g., kernel machines or deep neural nets), the Gaussian average is of order $\mathcal{O}(\sqrt{nK})$. Additionally, we define the normalized representation norm as:

$$\sup_{\Phi \in \mathcal{H}} \frac{1}{n\sqrt{K}} \|\Phi(\bar{\mathbf{X}})\| = \frac{1}{\sqrt{n}} \sup_{\Phi \in \mathcal{G}} \sqrt{\frac{1}{nK} \sum_{dki} \Phi_d(X_{ki})^2}.$$

This task-averaged perspective allows us to derive generalization bounds that reflect both empirical error and capacity control, setting the stage for our main theoretical results.

## 4 MAIN RESULTS

### 4.1 CROME WORKFLOW

To estimate the treatment effect on composite outcome, we propose CROME, which is a principled multi-task learning algorithm that jointly models multiple component outcomes using a shared representation. The central idea is to learn an encoder $\Phi$ that maps raw covariates $x \in \mathcal{X}$ to a shared latent space, from which outcome-specific predictors $f_k^{(a)}$ estimate the potential outcomes $Y_k(a)$ under each treatment arm $a \in \{0, 1\}$. These predictions are then aggregated through a user-specified utility function $u(\cdot)$ to obtain the composite potential outcomes $U(a)$. The final treatment effect estimate is computed as the difference $\hat{\tau}(x) = \hat{U}^{(1)}(x) - \hat{U}^{(0)}(x)$. The training procedure minimizes the empirical loss over all observed component outcomes using a multi-task architecture with parameter sharing across tasks.

The overall workflow of CROME is illustrated in Figure 2. This approach stands in contrast to single-task models, which directly regress the composite outcome on covariates and often fail to capture the structured relationships among individual components. The detailed steps of the proposed method are outlined in Algorithm 1 in Appendix B. The algorithm combines modular prediction of component-level outcomes with flexible utility-based aggregation, enabling accurate and interpretable estimation of treatment effects on composite outcomes.

### 4.2 GENERALIZATION ERROR BOUND

We then provide theoretical guarantees on the generalization performance of CROME. In particular, we analyze how representation sharing affects the treatment effect estimation error, and establish conditions under which our approach enjoys tighter generalization bounds compared to both independent task learning and direct composite prediction.

**Theorem 1** (Generalization Bound of CROME). *Let each potential outcome $\hat{Y}_k^{(a)}$ be modeled as $\hat{Y}_k^{(a)}(x) = f_k^{(a)}(\Phi(x))$, where $\Phi \in \mathcal{G}$ is a shared representation function, $f_k^{(a)} \in \mathcal{H}_k$ is the head for the $k$-th outcome under treatment $a \in \{0, 1\}$, and let the composite outcome be defined as $U(a) = u(Y_1(a), \ldots, Y_K(a))$, where $u$ is $L_u$-Lipschitz. Then, with probability at least $1 - \delta$ over a sample of size $n$, the expected treatment effect estimation error satisfies:*

$$\mathbb{E}_X[\ell(\hat{\tau}(X), \tau(X))]$$

$$\leq L_u \cdot \left(\sum_{k=1}^{K}\left(\mathcal{E}_k^{(0)} + \mathcal{E}_k^{(1)}\right) + \frac{2c_1 L G(\mathcal{H}(\bar{\mathbf{X}}))}{n} + \frac{2c_2 Q\sqrt{K}}{n} \sup_{\Phi \in \mathcal{G}} \|\Phi(\bar{\mathbf{X}})\| + \sqrt{\frac{32K\log(4/\delta)}{n}}\right),$$

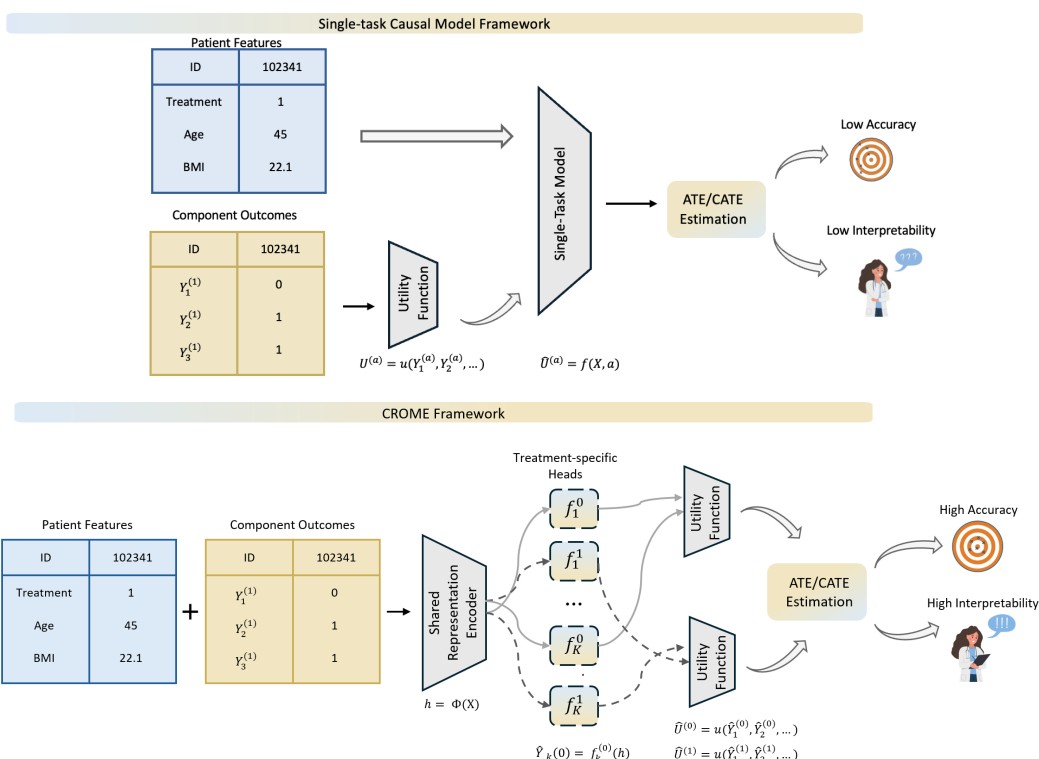

Figure 1: Comparison of multi-task representation learning approach versus single-task learning approach workflows for estimating treatment effects on composite outcomes. **Top:** The **single-task model** directly learns to predict the composite outcome without leveraging individual outcome structures. This approach may suffer from lower interpretability and degraded accuracy, especially when component outcomes contain informative and complementary signals. **Bottom:** Our proposed **CROME** learns a shared representation from patient covariates and predicts each individual outcome component via outcome-specific heads. The composite outcome is then constructed via a utility function over the predicted components, enabling interpretable and accurate estimation of treatment effects. The multi-task approach explicitly exploits this structure, leading to better performance.

where $\mathcal{E}_k^{(a)}$ is the empirical risk for outcome $k$ under treatment $a$, $c_1$ and $c_2$ are universal constants, and $Q$ is the quantity

$$Q = \sup_{y \neq y'} \frac{1}{\|y - y'\|} \mathbb{E} \sup_{f \in \mathcal{H}_k} \sum_{i=1}^{n} \gamma_i (f(y_i) - f(y_i')).$$

We provide the proof of Theorem 1 in Appendix A. This result offers a generalization bound for estimating treatment effects on the composite outcome within a multi-task learning framework that employs a shared representation. The central insight is that, when the composite outcome is defined by an $L_u$-Lipschitz utility function over multiple potential outcomes, the error in estimating treatment effects can be tightly controlled by ensuring accurate prediction of each component outcome. By leveraging task-specific prediction heads $f_k^{(a)}$ applied to a common representation $\Phi(x)$, the model enables parameter sharing across tasks and reduces overall model complexity. The generalization bound in Theorem 1 highlights two principal contributors to the expected treatment effect error: (i) the sum of empirical risks $\sum_k \mathcal{E}_k^{(a)}$ across all outcomes and treatment arms, and (ii) the complexity terms associated with the function classes used to learn the shared encoder and outcome-specific heads. These complexity terms include a Gaussian complexity term $G(\mathcal{H}(\bar{\mathbf{X}}))$, a spectral norm term $\sup_\Phi \|\Phi(\bar{\mathbf{X}})\|$, and a log-based confidence penalty $\sqrt{\log(1/\delta)/n}$.

To better understand the implications of Theorem 1, we next consider two propositions that formalize how shared representation learning compares to alternative modeling strategies. Proposition 1 shows that, under a mild complexity condition, CROME achieves a strictly tighter generalization bound than MTL without shared representation. Proposition 2 further demonstrates that the same approach can outperform single-task models that directly predict the composite outcome, which typically fail to leverage structure among the components and require larger function classes.

**Proposition 1** (CROME vs. MTL without Shared Representation). *Under the conditions of Theorem 1, consider an alternative model where each potential outcome is estimated independently as $\hat{Y}_k^{(a)}(x) = f_k^{(a)}(x)$, where $f_k^{(a)} \in \mathcal{F}_k$. Then, with probability at least $1 - \delta$,*

$$\mathbb{E}_X[\ell(\hat{\tau}(X), \tau(X))] \leq L_u \cdot \left( \sum_{k=1}^{K} \left( \mathcal{E}_k^{(1)} + \mathcal{E}_k^{(0)} + \mathcal{R}_n(\mathcal{F}_k) \right) + \sqrt{\frac{32 \log(4/\delta)}{n} K} \right).$$

*If the following inequality holds:*

$$\frac{c_1 L G(\mathcal{H}(\bar{\mathbf{X}}))}{n} + \frac{c_2 Q \sqrt{K}}{n} \sup_{\Phi \in \mathcal{G}} \|\Phi(\bar{\mathbf{X}})\| < \sum_{k=1}^{K} \mathcal{R}_n(\mathcal{F}_k)$$

*then the generalization bound in Theorem 1 is strictly tighter than that of the independent model.*

We provide the proof of Proposition 1 in Appendix A. This result formally establishes that the generalization bound for the shared representation model can be strictly tighter than that of an independent modeling strategy, under a realistic and theoretically grounded condition. The essential insight is that learning with shared representations allows for parameter sharing across multiple outcome prediction tasks, which reduces the overall function class complexity. In the independent modeling setting, each outcome-specific predictor is learned using a separate hypothesis class $\mathcal{F}_k$, leading to a total complexity that grows linearly with the number of outcomes: $\sum_{k=1}^{K} \mathcal{R}_n(\mathcal{F}_k)$. In contrast, the shared representation model factorizes the hypothesis space into a common encoder $\Phi \in \mathcal{G}$ and lightweight task-specific heads $f_k^{(a)} \in \mathcal{H}_k$, resulting a Gaussian complexity $G(\mathcal{H}(\bar{\mathbf{X}}))$ and the spectral norm term.

This condition is plausible in many settings because it compares the complexity of a shared multi-task model (left-hand side) with that of training $K$ independent models (right-hand side). The Gaussian complexity $G(\mathcal{H}(\bar{\mathbf{X}}))$ and the spectral norm term scale sub-linearly with $K$, typically as $\mathcal{O}(\sqrt{K})$, due to shared parameters and regularized representation learning. In contrast, the total Rademacher complexity of $K$ separate models grows linearly as $\mathcal{O}(K)$, since each task introduces a new set of parameters. Therefore, especially in settings with moderate or large $K$ and correlated outcomes—such as in healthcare or multi-label prediction—the shared representation model can achieve strictly lower overall complexity.

Next, we compare CROME with single-task learning approaches that directly predict the composite outcome. Let $g^{(a)} \in \mathcal{H}_{\text{comp}}$ denote a single-task model trained to estimate the composite potential outcome under treatment arm $a \in \{0, 1\}$. The empirical risk of this model under treatment $a$ is given by:

$$\mathcal{E}_{\text{comp}}^{(a)} := \frac{1}{n} \sum_{i=1}^{n} \ell\left( g^{(a)}(x_i), U_i(a) \right),$$

where $U_i(a) = u(Y_{i1}(a), \ldots, Y_{iK}(a))$ is the true composite outcome for unit $i$. The estimated treatment effect from this single-task model is then defined as:

$$\hat{\tau}_{\text{comp}}(x) := g^{(1)}(x) - g^{(0)}(x),$$

where $g^{(1)}(x)$ and $g^{(0)}(x)$ are the model's predictions for the treated and control conditions, respectively.

**Proposition 2** (CROME vs. Single-Task Learning). *Under the assumptions of Theorem 1, consider a single-task model that directly predicts the composite outcome via $\hat{U}^{(a)}(x) = g^{(a)}(x)$, where $g^{(a)} \in \mathcal{H}_{\text{comp}}$. Then, with probability at least $1 - \delta$,*

$$\mathbb{E}_X[\ell(\hat{\tau}_{\text{comp}}(X), \tau(X))] \leq \mathcal{E}_{\text{comp}}^{(1)} + \mathcal{E}_{\text{comp}}^{(0)} + 2 \cdot \mathcal{R}_n(\mathcal{H}_{\text{comp}}) + \sqrt{\frac{32 \log(4/\delta)}{n}}.$$

*If the composite utility function* u *is Boolean composite function and the following inequality holds:*

$$\frac{c_1 LG(\mathcal{H}(\bar{\mathbf{X}}))}{n} + \frac{c_2 Q \sqrt{K}}{n} \sup_{\Phi \in \mathcal{G}} \|\Phi(\bar{\mathbf{X}})\| \ll \mathcal{R}_n(\mathcal{H}_{comp}),$$

*then the shared multi-task model from Theorem 1 yields a strictly tighter generalization bound than the single-task composite model.*

We provide the proof of Proposition 2 in Appendix A. The condition in Proposition 2 captures the intuition that CROME can yield significantly lower generalization complexity than directly modeling the composite outcome. The left-hand side reflects the complexity of learning a shared encoder $\Phi$ and $K$ lightweight task-specific heads, while the right-hand side measures the complexity of a single-task model tasked with approximating the full composite mapping $x \mapsto U(a)$ in one shot. Since the composite outcome function implicitly entangles all $K$ outcomes through a potentially nonlinear utility function $u$, the hypothesis class $\mathcal{H}_{comp}$ must be expressive enough to capture all interactions—resulting in a much larger Rademacher complexity. In contrast, the shared representation model decomposes this problem into simpler subproblems, each with more tractable complexity, while leveraging parameter sharing. Therefore, the inequality holds in many practical settings, especially when $K$ is moderate to large or when component outcomes are conditionally dependent.

### 4.3 Interpretability via Decomposition of Treatment Effects on Composite Outcome

An important advantage of CROME is its ability to provide interpretable insights into how each component outcome contributes to the treatment effect on composite outcome. Specifically, by modeling each individual outcome $Y_k(a)$ separately, we can attribute the estimated treatment effect on the composite outcome $\hat{\tau}(x)$ to the contributions from each component outcome. To interpret the treatment effect on composite outcome, we consider a first-order approximation based on the Lipschitz property or differentiability of $u(\cdot)$. If $u$ is differentiable, then using Taylor expansion we approximate:

$$\hat{\tau}(x) \approx \sum_{k=1}^{K} \frac{\partial u}{\partial y_k}\Big|_{\bar{y}} \cdot \left( \hat{Y}_k^{(1)}(x) - \hat{Y}_k^{(0)}(x) \right),$$

where $\bar{y}$ is an intermediate point between the two predicted vectors. This decomposition expresses the treatment effect on composite outcome as a sum of component-level treatment effects, weighted by the sensitivity of the utility function to each outcome. In practice, this decomposition enables visualization of individual-level component contributions using tools such as heatmaps, or stacked bar charts (See Section 5.2). These tools can illustrate not only the magnitude but also the direction (positive or negative) of each component's influence on the treatment effect on composite outcome. Such interpretability is especially valuable in healthcare, where clinicians need to understand which clinical events are driving the overall effect and why a treatment may or may not be beneficial for a given patient.

## 5 Experiment

### 5.1 Treatment Effect Estimation on Composite Outcome

We evaluate our method across three benchmark synthetic and semi-synthetic datasets designed to test different aspects of treatment effect estimation on composite outcome. The data description and data-generating processes for all three datasets are described in detail in Appendix D.

**Synthetic Data.** We generate synthetic datasets with nonlinear shared latent structures among 4 component binary outcomes. Those 4 component outcomes are then composed into a binary composite outcome using the Boolean OR utility function.

**IHDP.** The Infant Health and Development Program (IHDP) (Hill, 2011) dataset is a standard semi-synthetic benchmark based on an observational study of early interventions for low-birthweight infants, with real covariates and treatment assignments. We generate 4 binary component outcomes given the covariate and treatment assignment and aggregate them using the Boolean OR utility function to construct a composite outcome.

**Cancer EHR.** The cancer EHR dataset is constructed from de-identified real-world clinical features collected from breast cancer patients at Penn Medicine. The treatment of interest is the chemotherapy. We simulate 4 binary patient-reported outcomes including fatigue, pain interference, anxiety, sleep disturbance. Then we aggregate them into a binary composite outcome using the Boolean OR utility function.

**Baselines.** We compare CROME against a comprehensive set of state-of-the-art baselines for treatment effect estimation, organized into the following categories. Tree-based methods include Causal Forest (Athey & Wager, 2019) and BART (Chipman et al., 2010). Double machine learning and doubly robust learners comprise DML (Chernozhukov et al., 2018) and DR-Learner (Kennedy, 2023). Meta-learners include S-Learner, T-Learner, and X-Learner (Künzel et al., 2019). Neural network-based causal representation learning models cover TARNet (Shalit et al., 2017), CFR-Net (Shalit et al., 2017), and DragonNet (Shi et al., 2019). Causal multi-task learning methods include CMGP (Alaa & Van Der Schaar, 2017) and CMDE (Jiang et al., 2023). Additionally, we include two composite-specific baseline variants: Single-Task Comp, which directly predicts the composite endpoint, and Multi-Task No Rep, which models component outcomes separately without shared representation. All baselines are extended to the multi-outcome setting by adding one output head for every (outcome, treatment) pair.

**Experimental Settings.** For all experiments, we trained CROME using an 80%/10%/10% split of the data into training, validation, and test sets, respectively. Hyperparameters were selected via a grid search over hidden layer widths $\{32, 64, 128\}$ and learning rates $\{10^{-3}, 10^{-4}\}$, choosing the configuration that minimized validation loss on the composite outcome. Models were trained using the Adam optimizer with early stopping based on validation loss, and a maximum of 200 training epochs. All experiments were implemented in Python 3.9.13 and conducted on a machine equipped with an Intel Xeon CPU (24 cores, 32 GB RAM) and an NVIDIA GeForce RTX 3080 GPU (10 GB VRAM), using CUDA 12.6 and NVIDIA driver version 560.94. Additional environment details are provided in Appendix F.

**Primary Results.** Table 1 reports the average absolute estimation error ($|\hat{\tau} - \tau|$) with standard errors across 1,000 simulation runs on synthetic, IHDP, and Cancer EHR datasets. CROME consistently achieves the lowest estimation error across all settings. On synthetic data, CROME attains the lowest in-sample error ($0.0020 \pm 0.0002$) and out-of-sample error ($0.0107 \pm 0.0002$), outperforming strong baselines including CMGP and DML. On IHDP, CROME achieves $0.0052 \pm 0.0001$ in-sample and $0.0108 \pm 0.0001$ out-of-sample error, again improving upon both classical and neural methods such as TARNet, DragonNet, and CMGP. The advantage persists in the EHR setting, where CROME reaches $0.0071 \pm 0.0001$ in-sample and $0.0107 \pm 0.0002$ out-of-sample error. Compared to the single-task composite model and the multi-task model without representation sharing, CROME yields substantial improvements. These results highlight the benefit of CROME for accurate and reliable estimation of treatment effects on composite endpoints. Additional experimental results and ablation studies—varying the number of component outcomes and the choice of utility function—together with runtime analyses are provided in Appendix E.

| Model | Synthetic Data | | IHDP | | Cancer EHR | |
|---|---|---|---|---|---|---|
| | In-sample Error | Out-of-sample Error | In-sample Error | Out-of-sample Error | In-sample Error | Out-of-sample Error |
| *TARnet* | $0.0046 \pm 0.0020$ | $0.0198 \pm 0.0037$ | $0.0063 \pm 0.0030$ | $0.0147 \pm 0.0027$ | $0.0112 \pm 0.0083$ | $0.0112 \pm 0.0083$ |
| *CFRnet* | $0.0330 \pm 0.0081$ | $0.0371 \pm 0.0068$ | $0.0362 \pm 0.0295$ | $0.1393 \pm 0.0281$ | $0.0135 \pm 0.0104$ | $0.1216 \pm 0.0122$ |
| *Dragonnet* | $0.0257 \pm 0.0042$ | $0.0088 \pm 0.0007$ | $0.0391 \pm 0.0138$ | $0.0653 \pm 0.0403$ | $0.0659 \pm 0.0381$ | $0.0680 \pm 0.0867$ |
| *BART* | $0.0117 \pm 0.0002$ | $0.0145 \pm 0.0003$ | $0.0144 \pm 0.0002$ | $0.0629 \pm 0.0002$ | $0.0351 \pm 0.0001$ | $0.0690 \pm 0.0001$ |
| *Causal Forest* | $0.0160 \pm 0.0001$ | $0.0305 \pm 0.0002$ | $0.0136 \pm 0.0002$ | $0.0151 \pm 0.0003$ | $0.0131 \pm 0.0002$ | $0.0143 \pm 0.0003$ |
| *DML* | $0.0051 \pm 0.0003$ | $0.0151 \pm 0.0001$ | $0.0078 \pm 0.0002$ | $0.0260 \pm 0.0003$ | $0.0384 \pm 0.0002$ | $0.0512 \pm 0.0001$ |
| *DR-Learner* | $0.0096 \pm 0.0002$ | $0.0230 \pm 0.0001$ | $0.0046 \pm 0.0002$ | $0.0311 \pm 0.0001$ | $0.0359 \pm 0.0002$ | $0.0107 \pm 0.0001$ |
| *S-Learner* | $0.0080 \pm 0.0003$ | $0.0119 \pm 0.0003$ | $0.0174 \pm 0.0002$ | $0.0139 \pm 0.0002$ | $0.0114 \pm 0.0001$ | $0.0141 \pm 0.0002$ |
| *T-Learner* | $0.0124 \pm 0.0002$ | $0.0237 \pm 0.0002$ | $0.0108 \pm 0.0002$ | $0.0356 \pm 0.0003$ | $0.0152 \pm 0.0002$ | $0.0168 \pm 0.0001$ |
| *X-Learner* | $0.0142 \pm 0.0001$ | $0.0318 \pm 0.0003$ | $0.0127 \pm 0.0003$ | $0.0281 \pm 0.0003$ | $0.0134 \pm 0.0001$ | $0.0194 \pm 0.0002$ |
| *CMDE* | $0.0164 \pm 0.0057$ | $0.0672 \pm 0.0210$ | $0.0089 \pm 0.0067$ | $0.1036 \pm 0.0073$ | $0.0100 \pm 0.0065$ | $0.1197 \pm 0.0071$ |
| *CMGP* | $0.0043 \pm 0.0013$ | $0.0681 \pm 0.0028$ | $0.0225 \pm 0.0029$ | $0.1217 \pm 0.0016$ | $0.0185 \pm 0.0026$ | $0.1202 \pm 0.0074$ |
| *Single-Task Comp* | $0.0073 \pm 0.0001$ | $0.0261 \pm 0.0002$ | $0.0059 \pm 0.0001$ | $0.0236 \pm 0.0001$ | $0.0208 \pm 0.0004$ | $0.0248 \pm 0.0005$ |
| *Multi-Task No Rep* | $0.0034 \pm 0.0003$ | $0.0128 \pm 0.0002$ | $0.0046 \pm 0.0002$ | $0.0162 \pm 0.0002$ | $0.0081 \pm 0.0001$ | $0.0185 \pm 0.0002$ |
| ***CROME*** | $\mathbf{0.0020 \pm 0.0002}$ | $\mathbf{0.0107 \pm 0.0002}$ | $\mathbf{0.0052 \pm 0.0001}$ | $\mathbf{0.0108 \pm 0.0001}$ | $\mathbf{0.0071 \pm 0.0001}$ | $\mathbf{0.0107 \pm 0.0002}$ |

Table 1: Average absolute estimation error ($|\hat{\tau} - \tau|$) across synthetic, IHDP, and Cancer EHR datasets.

## 5.2 Interpretability via Component-Wise Decomposition

Beyond accuracy, interpretability is crucial in clinical applications, particularly when working with complex EHR data. CROME enables decomposition of the treatment effect on composite outcome into contributions from individual outcome components, providing actionable insights into which outcome components are most affected by treatment at both the individual and population levels.

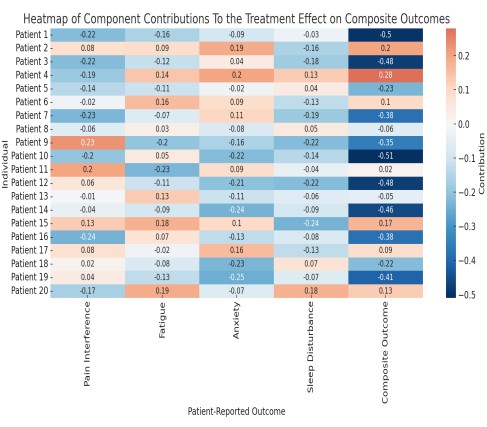
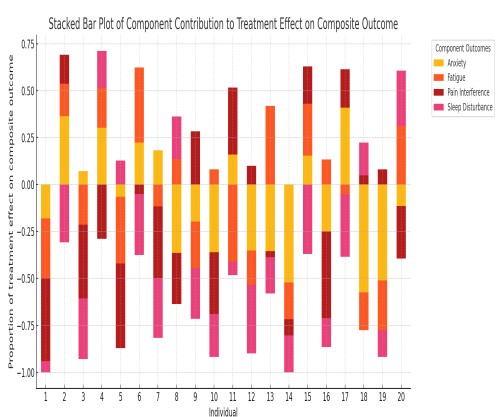

(a) Heatmap of component contributions to treatment effect on composite outcome.

(b) Stacked bar plot of normalized component contributions across individuals.

Figure 2: Visualization of component-level treatment effect contributions.

To illustrate our framework's interpretability, we visualize the component-level treatment effect contributions using two complementary tools: a heatmap and a stacked bar plot. The **heatmap** (Figure 2a) displays the contribution of 4 PROs—namely Pain Interference, Fatigue, Anxiety, and Sleep Disturbance—to the overall treatment effect on the composite outcome for individual patients. Each row corresponds to a patient, and columns represent the four component outcomes (first four columns) and the composite outcome (final column). Cell colors encode both the magnitude and direction of each component's contribution: warmer (red) tones indicate positive effects (i.e., treatment alleviates symptoms), while cooler (blue) tones indicate negative effects (i.e., treatment worsens symptoms). Within the Cancer EHR context, red signifies symptom improvement (e.g., reductions in Pain Interference or Fatigue), whereas blue suggests symptom deterioration. This visualization enables granular, patient-level interpretation and reveals heterogeneous treatment response patterns. For instance, some patients benefit primarily through reductions in fatigue and sleep disturbance, while others exhibit limited improvements in physical or emotional functioning.

The **stacked bar plot** (Figure 2b) visualizes how each component outcome contributes to the treatment effect on composite outcome at the individual level. Each bar represents a single patient, and all bars are normalized to have the same total height, enabling direct comparison across individuals. The segments within each bar indicate the relative contribution of each patient-reported outcome—Pain Interference, Fatigue, Anxiety, and Sleep Disturbance—such that their proportions sum to one. Negative contributions (below the x-axis) indicate that a component outcome worsened due to treatment, while positive contributions (above the x-axis) indicate improvement. This visualization reveals heterogeneous response patterns: for example, some individuals show strong improvement primarily through reductions in fatigue and sleep disturbance, while others display mixed effects across outcomes. This plot complements the heatmap by offering a holistic, compositional view of how treatment benefits (or harms) are distributed across multiple outcomes for each patient.

## 6 Discussion and Conclusion

**CROME** opens new directions for treatment effect estimation in settings with complex, multi-dimensional outcome structures, particularly in healthcare where composite outcomes are common. By enabling interpretable, component-level decomposition of treatment effects and supporting

flexible utility-based aggregation, CROME offers a principled approach for personalized decision-making. Its strong empirical performance and generalizability make it a promising tool for broader applications beyond healthcare. Future extensions could incorporate time-varying treatments, longitudinal data, and domain-adaptive learning for multi-center EHR systems.

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

# A PROOFS OF MAINS RESULTS

To establish our main theoretical results, we introduce a set of analytical tools that connect multi-task prediction accuracy with bounds on treatment effect estimation error. These tools are central to proving Theorem 1, which provides generalization guarantees for estimating treatment effects on composite outcome under a shared representation learning framework. Specifically, we leverage results from statistical learning theory—particularly bounds involving empirical risk and complexity measures such as Rademacher and Gaussian averages—to control the excess risk incurred by learning both the shared encoder and the task-specific heads.

Building on this foundation, Proposition 2 and Proposition 3 formalize the advantages of the shared representation model over two natural baselines: (i) independent models that learn each outcome task separately, and (ii) a single-task model that directly predicts the composite outcome. By combining Lipschitz continuity of the utility function with generalization theory for multi-task learning, we show that our proposed approach achieves strictly tighter error bounds under reasonable assumptions about the complexity of the function classes involved.

For convenience, we summarize all key mathematical symbols and notation used throughout the paper in Table 2.

| Symbol | Description |
|---|---|
| $\mathcal{X}$ | Covariate space; $x \in \mathcal{X}$ denotes an input (e.g., EHR features). |
| $A \in \{0, 1\}$ | Binary treatment indicator; $A = 1$ for treatment, $A = 0$ for control. |
| $Y_k(a) \in \mathbb{R}$ | Potential outcome for component $k$ under treatment $a$. |
| $\hat{Y}_k^{(a)}(x)$ | Predicted potential outcome for component $k$ under treatment $a$. |
| $U(a)$ | Composite outcome under treatment $a$, defined as $u(Y_1(a), \ldots, Y_K(a))$. |
| $\hat{U}^{(a)}(x)$ | Predicted composite outcome, $u(\hat{Y}_1^{(a)}(x), \ldots, \hat{Y}_K^{(a)}(x))$. |
| $\tau(x)$ | Conditional average treatment effect: $\mathbb{E}[U(1) - U(0) \mid x]$. |
| $\hat{\tau}(x)$ | Estimated CATE: $\hat{U}^{(1)}(x) - \hat{U}^{(0)}(x)$. |
| $\hat{\tau}_{\text{comp}}(x)$ | Estimated CATE using a single-task model trained on the composite outcome. |
| $u$ | Utility function combining $K$ component outcomes; assumed $L_u$-Lipschitz w.r.t. $\ell_1$. |
| $\mathcal{E}_k^{(a)}$ | Empirical loss for predicting $Y_k(a)$. |
| $\mathcal{E}_{\text{comp}}^{(a)}$ | Empirical loss for predicting the composite outcome $U(a)$ using a single-task model. |
| $\mathcal{F}_k$ | Hypothesis class for outcome $Y_k(a)$ with independent modeling. |
| $\mathcal{H}_k$ | Hypothesis class for the task-specific head for outcome $k$. |
| $\mathcal{G}$ | Hypothesis class for shared representation function $\Phi(x)$. |
| $\mathcal{F}_u$ | Hypothesis class for directly modeling the composite outcome $U(a)$. |
| $\mathcal{R}_n(\mathcal{F})$ | Empirical Rademacher complexity of class $\mathcal{F}$. |
| $\hat{\mathcal{G}}_S(\mathcal{F})$ | Empirical Gaussian complexity of class $\mathcal{F}$. |
| $\ell(\cdot, \cdot)$ | Loss function used for training (e.g., cross-entropy, squared loss). |
| $\Phi(x)$ | Shared representation of input $x$, $\Phi : \mathcal{X} \to \mathbb{R}^D$. |
| $f_k^{(a)}$ | Prediction head for component $k$ under treatment $a$, mapping $\mathbb{R}^D \to [0, 1]$. |
| $\mathcal{R}_{\text{bal}}$ | Balance regularizer which quantifies how different the learned representation distributions are between the treated and control groups |

Table 2: Summary of key notation used throughout the paper.

**Lemma 1** (Treatment Effect Error Bound via Component Prediction). *Let $U(a) = u(Y_1(a), \ldots, Y_K(a))$ denote a composite outcome defined via a utility function $u : \mathbb{R}^K \to \mathbb{R}$, and let $\hat{U}^{(a)}(x) = u(\hat{Y}_1^{(a)}(x), \ldots, \hat{Y}_K^{(a)}(x))$ be the predicted composite outcome. Assume that $u$ is $L_u$-Lipschitz with respect to the $\ell_1$ norm. Then for all $x \in \mathcal{X}$,*

$$\ell\left(\hat{\tau}(x), \tau(x)\right) \le L_u \cdot \sum_{k=1}^{K} \left( \ell(\hat{Y}_k^{(1)}(x), \mathbb{E}[Y_k(1) \mid x]) + \ell(\hat{Y}_k^{(0)}(x), \mathbb{E}[Y_k(0) \mid x]) \right)$$

*where $\tau(x) = \mathbb{E}[U(1) - U(0) \mid x]$ and $\hat{\tau}(x) = \hat{U}^{(1)}(x) - \hat{U}^{(0)}(x)$.*

*Proof.* Let $\tau(x) = \mathbb{E}[U(1) - U(0) \mid x]$ denote the true conditional treatment effect on the composite outcome, and let $\hat{\tau}(x) = \hat{U}^{(1)}(x) - \hat{U}^{(0)}(x)$ be the estimated counterpart. We aim to bound the discrepancy $\ell(\hat{\tau}(x), \tau(x))$ using the prediction errors for the individual component outcomes.

By the triangle inequality for general loss functions (e.g., when $\ell$ is convex and satisfies the triangle inequality), we have:

$$\ell(\hat{\tau}(x), \tau(x)) = \ell\left(\hat{U}^{(1)}(x) - \hat{U}^{(0)}(x), \mathbb{E}[U(1) \mid x] - \mathbb{E}[U(0) \mid x]\right)$$
$$\le \ell(\hat{U}^{(1)}(x), \mathbb{E}[U(1) \mid x]) + \ell(\hat{U}^{(0)}(x), \mathbb{E}[U(0) \mid x]).$$

Now consider each term of the form $\ell(\hat{U}^{(a)}(x), \mathbb{E}[U(a) \mid x])$ for $a \in \{0, 1\}$. Since the composite outcome is defined as $U(a) = u(Y_1(a), \ldots, Y_K(a))$ and its estimate is $\hat{U}^{(a)}(x) = u(\hat{Y}_1^{(a)}(x), \ldots, \hat{Y}_K^{(a)}(x))$, and assuming $u$ is $L_u$-Lipschitz with respect to the $\ell_1$ norm, we apply the Lipschitz property:

$$\ell(\hat{U}^{(a)}(x), \mathbb{E}[U(a) \mid x]) = \ell\left(u(\hat{Y}_1^{(a)}(x), \ldots, \hat{Y}_K^{(a)}(x)), u\left(\mathbb{E}[Y_1(a) \mid x], \ldots, \mathbb{E}[Y_K(a) \mid x]\right)\right)$$
$$\le L_u \sum_{k=1}^{K} \ell\left(\hat{Y}_k^{(a)}(x), \mathbb{E}[Y_k(a) \mid x]\right).$$

Substituting back into the earlier bound:

$$\ell(\hat{\tau}(x), \tau(x)) \le L_u \sum_{k=1}^{K} \left( \ell\left(\hat{Y}_k^{(1)}(x), \mathbb{E}[Y_k(1) \mid x]\right) + \ell\left(\hat{Y}_k^{(0)}(x), \mathbb{E}[Y_k(0) \mid x]\right) \right),$$

which completes the proof. $\square$

**Corollary 1** (Boolean Composite Case). *If $U(a) = 1 - \prod_{k=1}^{K}(1 - Y_k(a))$, then $u$ is 1-Lipschitz and the bound becomes:*

$$|\hat{\tau}(x) - \tau(x)| \le \sum_{k=1}^{K} \left( \left| \hat{Y}_k^{(1)}(x) - \mathbb{E}[Y_k(1) \mid x] \right| + \left| \hat{Y}_k^{(0)}(x) - \mathbb{E}[Y_k(0) \mid x] \right| \right).$$

*Proof.* Let the composite outcome be defined as

$$U(a) = u(Y_1(a), \ldots, Y_K(a)) = 1 - \prod_{k=1}^{K}(1 - Y_k(a)),$$

which corresponds to the Boolean "OR" function on the outcomes (i.e., $U(a) = 1$ if any $Y_k(a) = 1$).

We want to verify that the utility function $u : [0, 1]^K \to [0, 1]$, given by

$$u(y_1, \ldots, y_K) = u(\boldsymbol{y}) = 1 - \prod_{k=1}^{K}(1 - y_k),$$

is 1-Lipschitz with respect to the $\ell_1$ norm where $\boldsymbol{y} = (y_1, \ldots, y_K)$.

To do this, we compute the partial derivative of $u$ with respect to each coordinate:

$$\frac{\partial u}{\partial y_k} = \prod_{j \ne k}(1 - y_j).$$

Since each $y_j \in [0, 1]$, it follows that $0 \leq \frac{\partial u}{\partial y_k} \leq 1$, and thus:

$$\|\nabla u(\boldsymbol{y})\|_1 = \sum_{k=1}^K \left| \frac{\partial u}{\partial y_k} \right| = \sum_{k=1}^K \prod_{j \neq k} (1 - y_j) \leq \sum_{k=1}^K 1 = K.$$

But more precisely, since $u$ is differentiable on $[0, 1]^K$, and each partial derivative is in $[0, 1]$, we have:

$$|u(\boldsymbol{y}) - u(\boldsymbol{y}')| \leq \sum_{k=1}^K \left| \frac{\partial u}{\partial y_k} \right| \cdot |y_k - y_k'| \leq \sum_{k=1}^K |y_k - y_k'| = \|\boldsymbol{y} - \boldsymbol{y}'\|_1.$$

Hence, $u$ is 1-Lipschitz with respect to the $\ell_1$ norm.

Now applying Lemma 1 with $L_u = 1$ and $\ell(a, b) = |a - b|$, we obtain:

$$|\hat{\tau}(x) - \tau(x)| \leq \sum_{k=1}^K \left( |\hat{Y}_k^{(1)}(x) - \mathbb{E}[Y_k(1) \mid x]| + |\hat{Y}_k^{(0)}(x) - \mathbb{E}[Y_k(0) \mid x]| \right),$$

as desired. $\qquad\square$

To assess representation quality, we define the task-averaged population risk under treatment $a$ as:

$$\mathcal{E}_{\mathrm{avg}}^{(a)}(\Phi, f_1, \ldots, f_K) = \frac{1}{K} \sum_{k=1}^K \mathbb{E}_{(X,Y) \sim \mu_k} \ell(f_k^{(a)}(\Phi(X)), \mathbb{E}[Y_k(a)|X]),$$

and the corresponding minimal achievable risk is:

$$\mathcal{E}_{\mathrm{avg}}^*(a) = \min_{\Phi \in \mathcal{G}, f_k \in \mathcal{H}_k} \mathcal{E}_{\mathrm{avg}}^{(a)}(\Phi, f_1, \ldots, f_K).$$

Having defined the relevant complexity measures and population risks, we are now positioned to analyze the generalization performance of the shared representation model. Specifically, we aim to bound the difference between the task-averaged population risk of the learned model and the best achievable risk within the hypothesis classes $\mathcal{G}$ and $\{\mathcal{H}_k\}_{k=1}^K$. The following result, adapted from Maurer et al. (2016), establishes a high-probability upper bound on the excess risk in terms of the Gaussian complexity of the representation class and the empirical norm of the induced feature maps. This result forms the foundation for our generalization analysis of treatment effect estimation on composite outcome.

**Theorem 2** (Maurer et al. (2016)). *Let $\mu_1, \cdots, \mu_K$ and $\mathcal{F}_k$ be as above, and assume $0 \in \mathcal{G}$ and $f(0) = 0$ for all $f \in \mathcal{F}_k$. Then for $\delta > 0$ with probability at least $1 - \delta$ in the draw of $\bar{\mathbf{Z}} \sim \Pi_{k=1}^K \mu_k^n$ we have*

$$\mathcal{E}_{avg}^{(a)}(\hat{\Phi}, \hat{f}_1, \cdots, \hat{f}_K) - \mathcal{E}_{avg}^*(a) \leq \frac{c_1 L G(\mathcal{H}(\bar{\mathbf{X}}))}{nK} + \frac{c_2 Q \sup_{\Phi \in \mathcal{G}} \|\Phi(\bar{\mathbf{X}})\|}{n\sqrt{K}} + \sqrt{\frac{8 \ln(4/\delta)}{nK}},$$

*where $c_1$ and $c_2$ are universal constants, and $Q$ is the quantity*

$$Q = \sup_{y \neq y'} \frac{1}{\|y - y'\|} \mathbb{E} \sup_{f \in \mathcal{H}_k} \sum_{i=1}^n \gamma_i (f(y_i) - f(y_i')).$$

The proof of Theorem 2 can be found in Maurer et al. (2016). We now provide the proof of Theorem 1.

*Proof of Theorem 1.* We begin by applying Lemma 1, which relates the treatment effect estimation error to the prediction errors on each potential outcome component via the Lipschitz property of the utility function $u$. Specifically, since $u$ is $L_u$-Lipschitz with respect to the $\ell_1$ norm, we have:

$$\ell(\hat{\tau}(x), \tau(x)) = \ell\left(\hat{U}^{(1)}(x) - \hat{U}^{(0)}(x), \mathbb{E}[U(1) - U(0) \mid x]\right) \leq L_u \sum_{k=1}^K \sum_{a=0}^1 \ell\left(\hat{Y}_k^{(a)}(x), \mathbb{E}[Y_k(a) \mid x]\right).$$

Taking the expectation over the covariate distribution $x \sim \mathcal{D}$ on both sides yields:

$$\mathbb{E}_X[\ell(\hat{\tau}(X), \tau(X))] \leq L_u \sum_{k=1}^{K} \sum_{a=0}^{1} \mathbb{E}_X\left[\ell\left(\hat{Y}_k^{(a)}(X), \mathbb{E}[Y_k(a)|X]\right)\right].$$

Now, under the model assumption that $\hat{Y}_k^{(a)}(x) = f_k^{(a)}(\Phi(x))$, we apply the generalization bound for shared representation multi-task learning due to Maurer et al. (2016). For each treatment arm $a \in \{0,1\}$, this bound implies that, with probability at least $1 - \delta$, the task-averaged expected loss satisfies:

$$\frac{1}{K} \sum_{k=1}^{K} \mathbb{E}_X[\ell(f_k^{(a)}(\Phi(X)), \mathbb{E}[Y_k(a)|X])] \leq \frac{1}{K} \sum_{k=1}^{K} \mathcal{E}_k^{(a)} + \frac{c_1 L G(\mathcal{H})}{nK} + \frac{c_2 Q \sup_{\Phi \in \mathcal{G}} \|\Phi(\bar{\mathbf{X}})\|}{n\sqrt{K}} + \sqrt{\frac{8 \log(4/\delta)}{nK}}.$$

Multiplying both sides by $K$ converts the average back into a full sum over tasks:

$$\sum_{k=1}^{K} \mathbb{E}_X[\ell(f_k^{(a)}(\Phi(X)), \mathbb{E}[Y_k(a)|X])] \leq \sum_{k=1}^{K} \mathcal{E}_k^{(a)} + \frac{c_1 L G(\mathcal{H})}{n} + c_2 Q \cdot \frac{\sqrt{K}}{n} \sup_{\Phi \in \mathcal{G}} \|\Phi(\bar{\mathbf{X}})\| + \sqrt{\frac{8K \log(4/\delta)}{n}}.$$

Now summing over both treatment arms $a \in \{0,1\}$:

$$\sum_{a=0}^{1} \sum_{k=1}^{K} \mathbb{E}_X[\ell(f_k^{(a)}(\Phi(X)), \mathbb{E}[Y_k(a)|X])]$$

$$\leq \sum_{k=1}^{K} \left(\mathcal{E}_k^{(0)} + \mathcal{E}_k^{(1)}\right) + \frac{2c_1 L G(\mathcal{H})}{n} + 2c_2 Q \cdot \frac{\sqrt{K}}{n} \sup_{\Phi \in \mathcal{G}} \|\Phi(\bar{\mathbf{X}})\| + 2 \cdot \sqrt{\frac{8K \log(4/\delta)}{n}}.$$

Finally, plugging this back into the earlier bound on treatment effect error, we obtain:

$$\mathbb{E}_X[\ell(\hat{\tau}(X), \tau(X))]$$

$$\leq L_u \cdot \left(\sum_{k=1}^{K} \left(\mathcal{E}_k^{(0)} + \mathcal{E}_k^{(1)}\right) + \frac{2c_1 L G(\mathcal{H})}{n} + \frac{2c_2 Q \sqrt{K}}{n} \sup_{\Phi \in \mathcal{G}} \|\Phi(\bar{\mathbf{X}})\| + \sqrt{\frac{32K \log(4/\delta)}{n}}\right).$$

This completes the proof of Theorem 1. $\qquad\qquad\square$

*Proof of Proposition 1.* From Theorem 1, the shared representation model satisfies, with probability at least $1 - \delta$:

$$\mathbb{E}_X[\ell(\hat{\tau}(X), \tau(X))]$$

$$\leq L_u \cdot \left(\sum_{k=1}^{K} \left(\mathcal{E}_k^{(0)} + \mathcal{E}_k^{(1)}\right) + \frac{2c_1 L G(\mathcal{H})}{n} + \frac{2c_2 Q \sqrt{K}}{n} \sup_{\Phi \in \mathcal{G}} \|\Phi(\bar{\mathbf{X}})\| + \sqrt{\frac{32K \log(4/\delta)}{n}}\right).$$

Now consider an independent modeling strategy, where each outcome $\hat{Y}_k^{(a)}$ is modeled using its own function $f_k^{(a)} \in \mathcal{F}_k$. Applying standard Rademacher generalization bounds for each $k$ and treatment $a \in \{0,1\}$, we have, with probability at least $1 - \delta$ over the training data:

$$\mathbb{E}_X\left[\ell(\hat{Y}_k^{(a)}(X), \mathbb{E}[Y_k(a)|X])\right] \leq \mathcal{E}_k^{(a)} + \mathcal{R}_n(\mathcal{F}_k) + \sqrt{\frac{8 \log(4/\delta)}{n}}.$$

Summing over all $k$ and both treatment arms and using the treatment effect error decomposition from Proposition 3 (the Lipschitz utility result), we obtain:

$$\mathbb{E}_X[\ell(\hat{\tau}(X), \tau(X))] \leq L_u \cdot \sum_{k=1}^{K} \left(\mathbb{E}_X\left[\ell(\hat{Y}_k^{(1)}(X), \mathbb{E}[Y_k(1)|X])\right] + \mathbb{E}_X\left[\ell(\hat{Y}_k^{(0)}(X), \mathbb{E}[Y_k(0)|X])\right]\right)$$

$$\leq L_u \cdot \sum_{k=1}^{K} \left(\mathcal{E}_k^{(1)} + \mathcal{E}_k^{(0)} + 2\mathcal{R}_n(\mathcal{F}_k) + \sqrt{\frac{32 \log(4/\delta)}{n}}\right)$$

$$= L_u \cdot \left(\sum_{k=1}^{K} \left(\mathcal{E}_k^{(1)} + \mathcal{E}_k^{(0)} + 2\mathcal{R}_n(\mathcal{F}_k)\right) + \sqrt{\frac{32 \log(4/\delta)}{n}} K\right).$$

Now comparing the two generalization bounds:

- The shared representation model has complexity:

$$\frac{2c_1 LG(\mathcal{H})}{n} + \frac{2c_2 Q\sqrt{K}}{n} \sup_{\Phi \in \mathcal{G}} \|\Phi(\bar{\mathbf{X}})\|$$

- The independent model has complexity:

$$2 \sum_{k=1}^{K} \mathcal{R}_n(\mathcal{F}_k)$$

Hence, if the following condition holds:

$$\frac{c_1 LG(\mathcal{H})}{n} + \frac{c_2 Q\sqrt{K}}{n} \sup_{\Phi \in \mathcal{G}} \|\Phi(\bar{\mathbf{X}})\| < \sum_{k=1}^{K} \mathcal{R}_n(\mathcal{F}_k),$$

then the generalization bound for the shared representation model is strictly tighter than that of the independent model, completing the proof. $\qquad\square$

*Proof of Proposition 2.* Let the single-task model directly predict the composite outcome using $g^{(a)} \in \mathcal{H}_{\text{comp}}$, so that:

$$\hat{U}^{(a)}(x) = g^{(a)}(x), \quad \text{and} \quad \hat{\tau}_{\text{comp}}(x) = g^{(1)}(x) - g^{(0)}(x).$$

The prediction error of the treatment effect on the composite outcome can be bounded using standard Rademacher-based generalization results:

$$\begin{aligned}
\mathbb{E}_X[\ell(\hat{\tau}_{\text{comp}}(X), \tau(X))] &= \mathbb{E}_X[\ell(g^{(1)}(X) - g^{(0)}(X), \tau(X))] \\
&\leq \mathbb{E}_X[\ell(g^{(1)}(X), U(1))] + \mathbb{E}_X[\ell(g^{(0)}(X), U(0))] \\
&\leq \mathcal{E}_{\text{comp}}^{(1)} + \mathcal{E}_{\text{comp}}^{(0)} + 2\mathcal{R}_n(\mathcal{H}_{\text{comp}}) + \sqrt{\frac{32 \log(4/\delta)}{n}},
\end{aligned}$$

where $\mathcal{E}_{\text{comp}}^{(a)}$ denotes the empirical loss under treatment $a$ for predicting the composite outcome:

$$\mathcal{E}_{\text{comp}}^{(a)} := \frac{1}{n} \sum_{i=1}^{n} \ell\left(g^{(a)}(x_i), U_i(a)\right),$$

and $\mathcal{R}_n(\mathcal{H}_{\text{comp}})$ is the empirical Rademacher complexity of the function class $\mathcal{H}_{\text{comp}}$ used to model the composite outcome directly.

Now, consider the multi-task shared representation approach as described in Theorem 1. That model predicts each component outcome $\hat{Y}_k^{(a)}(x) = f_k^{(a)}(\Phi(x))$, and the composite estimate is constructed as:

$$\hat{U}^{(a)}(x) = u\left(\hat{Y}_1^{(a)}(x), \dots, \hat{Y}_K^{(a)}(x)\right),$$

where $u$ is an $L_u$-Lipschitz utility function.

Theorem 1 guarantees that, with probability at least $1 - \delta$:

$$\begin{aligned}
\mathbb{E}_X&[\ell(\hat{\tau}(X), \tau(X))] \\
&\leq L_u \cdot \left( \sum_{k=1}^{K} \left( \mathcal{E}_k^{(0)} + \mathcal{E}_k^{(1)} \right) + \frac{2c_1 LG(\mathcal{H})}{n} + \frac{2c_2 Q\sqrt{K}}{n} \sup_{\Phi \in \mathcal{G}} \|\Phi(\bar{\mathbf{X}})\| + \sqrt{\frac{32K \log(4/\delta)}{n}} \right).
\end{aligned}$$

For a fixed number of outcomes $K$, the total empirical loss incurred by the multi-task model, $\sum_{k=1}^{K} \left( \mathcal{E}_k^{(0)} + \mathcal{E}_k^{(1)} \right)$, is often lower than the empirical loss for the single-task model trained directly on the composite outcome, $\mathcal{E}_{\text{comp}}^{(0)} + \mathcal{E}_{\text{comp}}^{(1)}$. This is because directly predicting the composite

outcome involves learning a more complex, often nonlinear and non-smooth mapping $u(\cdot)$, whereas the multi-task model only needs to fit simpler binary labels. Additionally, modeling each component outcome separately allows better exploitation of structure, leading to improved empirical performance and generalization. Since $L_u = 1$ for *Boolean composite function*, comparing the two bounds, we conclude that if:

$$\frac{c_1 LG(\mathcal{H})}{n} + \frac{c_2 Q\sqrt{K}}{n} \sup_{\Phi \in \mathcal{G}} \|\Phi(\bar{\mathbf{X}})\| \ll \mathcal{R}_n(\mathcal{H}_{\text{comp}}),$$

which is saying that the reduced complexity in the shared model can more than compensate for the larger $\sqrt{K}$ confidence term, then the multi-task shared representation model enjoys a strictly tighter generalization bound for estimating the treatment effect than the single-task composite predictor.
$\square$

## B  CROME ALGORITHM

Algorithm 1 outlines the full CROME procedure with two stages: *training* and *estimation*. During training, CROME jointly learns a shared representation $\Phi$ and task-specific heads $\{f_k^{(a)}\}$ to predict component-level potential outcomes under each arm. For each minibatch, the model computes $\Phi(x_i)$, applies the appropriate head to obtain predictions, and aggregates per-task losses into a single objective minimized via backpropagation. With a loss $\ell$ (e.g., cross-entropy or squared error) that is $L$-Lipschitz in $\ell_1$, the per-task empirical loss is

$$\mathcal{E}_k^{(a)} = \frac{1}{n_a} \sum_{i \in \mathcal{I}_a} \ell\Big(f_k^{(a)}\big(\Phi(x_i)\big), y_{ik}\Big), \quad k = 1, \dots, K, \ a \in \{0, 1\}.$$

To mitigate distributional imbalance between treated and control groups in the latent space, we add a *balance regularization* term that penalizes divergence between the distributions of $\Phi(X)$ for $A = 1$ and $A = 0$. Let $\mathcal{H}_1 = \{\Phi(x_i) : i \in \mathcal{I}_1\}$ and $\mathcal{H}_0 = \{\Phi(x_i) : i \in \mathcal{I}_0\}$, and define

$$\mathcal{R}_{\text{bal}} := D(\mathcal{H}_1, \mathcal{H}_0),$$

where $D$ can be instantiated as the MMD with a characteristic kernel $k$:

$$\text{MMD}^2(\mathcal{H}_1, \mathcal{H}_0) = \frac{1}{n_1^2} \sum_{i,i' \in \mathcal{I}_1} k\big(\Phi(x_i), \Phi(x_{i'})\big) + \frac{1}{n_0^2} \sum_{j,j' \in \mathcal{I}_0} k\big(\Phi(x_j), \Phi(x_{j'})\big) - \frac{2}{n_1 n_0} \sum_{i \in \mathcal{I}_1} \sum_{j \in \mathcal{I}_0} k\big(\Phi(x_i), \Phi(x_j)\big).$$

The training objective is the balance-augmented empirical loss

$$\mathcal{L}_{\text{total}} = \sum_{k=1}^{K} \big(\mathcal{E}_k^{(0)} + \mathcal{E}_k^{(1)}\big) + \lambda \mathcal{R}_{\text{bal}}, \quad \lambda > 0,$$

which trades off predictive accuracy and representation balance.

At the estimation stage, the learned $\Phi$ and $\{f_k^{(a)}\}$ produce component-level potential outcomes for a new input $x$, $\hat{Y}_k^{(a)}(x) = f_k^{(a)}(\Phi(x))$. A user-defined utility $u$ composes these into composite potential outcomes $\hat{U}^{(a)} = u\big(\hat{Y}_1^{(a)}(x), \dots, \hat{Y}_K^{(a)}(x)\big)$, and the individual-level treatment effect is

$$\hat{\tau}(x) = \hat{U}^{(1)} - \hat{U}^{(0)}.$$

This modular structure enables flexible and interpretable estimation of treatment effects on composite outcomes while promoting balance in the learned representation.

## C  INTERPRETABILITY VIA DECOMPOSITION OF TREATMENT EFFECT ON COMPOSITE OUTCOME: BOOLEAN UTILITY FUNCTION

A key advantage of our multi-task framework is its ability to provide interpretable estimates of treatment effects on composite outcome by decomposing them into contributions from individual

---

**Algorithm 1** CROME with Target Regularization

---

1: **Input:** Observed data $\{(x_i, a_i, y_{i1}, \ldots, y_{iK})\}_{i=1}^n$, Utility function $u : \mathbb{R}^K \to \mathbb{R}$, balancing regularization parameter $\lambda > 0$.
2: **Output:** $\hat{\tau}(x)$.
3: Initialize parameters of $\Phi$ and $\{f_k^{(a)}\}$
4: **for** each epoch **do**
5:     **for** each minibatch $\{(x_i, a_i, y_{i1}, \ldots, y_{iK})\}$ **do**
6:         Compute shared representations $h_i = \Phi(x_i)$
7:         Separate treated and control representations:

$$\mathcal{H}_1 = \{h_i \mid a_i = 1\}, \quad \mathcal{H}_0 = \{h_i \mid a_i = 0\}$$

8:         **for** each task $k = 1, \ldots, K$ **do**
9:             Predict outcome $\hat{y}_{ik}^{(a_i)} = f_k^{(a_i)}(h_i)$
10:           Compute loss $\ell_k = \ell(\hat{y}_{ik}^{(a_i)}, y_{ik})$
11:         **end for**
12:         Compute target regularization loss (e.g., MMD or Wasserstein distance):

$$\mathcal{R}_{\text{bal}} = \text{BalanceLoss}(\mathcal{H}_1, \mathcal{H}_0)$$

13:         Compute total loss:

$$\mathcal{L} = \sum_{k=1}^{K} \ell_k + \lambda \cdot \mathcal{R}_{\text{bal}}$$

14:         Update all parameters to minimize $\mathcal{L}$
15:     **end for**
16: **end for**
17: **Estimation of Treatment Effects:**
18: **for** each test point $x$ **do**
19:     $h = \Phi(x)$
20:     $\hat{y}_k^{(a)} = f_k^{(a)}(h)$ for all $k$ and $a \in \{0, 1\}$
21:     $\hat{U}^{(a)} = u(\hat{y}_1^{(a)}, \ldots, \hat{y}_K^{(a)})$
22:     $\hat{\tau}(x) = \hat{U}^{(1)} - \hat{U}^{(0)}$
23: **end for**

---

outcome components. This interpretability is crucial in clinical applications where understanding which specific outcomes drive treatment effects can inform patient-level decision making.

We choose the Boolean OR utility function as the default composite outcome in our experiments because it aligns with a common clinical decision-making criterion: treatment is considered beneficial if it improves at least one of several key outcomes. This formulation captures the intuition behind many composite endpoints used in practice, such as reducing any among multiple symptoms (e.g., fatigue, pain, or anxiety) in patient-reported outcomes. Additionally, the OR function is simple, interpretable, and provides a natural testbed to validate whether the model can effectively capture non-additive dependencies among component outcomes.

Let $Y_k(a) \in \{0, 1\}$ denote the $k$-th binary potential outcome under treatment $a \in \{0, 1\}$, and $\hat{Y}_k^{(a)}(x) \in [0, 1]$ be the model's predicted outcome. We define the composite outcome under treatment $a$ using the Boolean OR utility function:

$$U(a) = 1 - \prod_{k=1}^{K} (1 - Y_k(a)), \quad \hat{U}^{(a)}(x) = 1 - \prod_{k=1}^{K} \left(1 - \hat{Y}_k^{(a)}(x)\right).$$

This function equals 1 if at least one $Y_k(a) = 1$, and 0 otherwise. The estimated treatment effect on the composite outcome is:

$$\hat{\tau}(x) = \hat{U}^{(1)}(x) - \hat{U}^{(0)}(x).$$

To interpret $\hat{\tau}(x)$ in terms of component contributions, we apply a first-order Taylor expansion to the function $u(y) = 1 - \prod_{k=1}^{K}(1 - y_k)$ around a reference point $\bar{y} = (\bar{y}_1, \ldots, \bar{y}_K)$ (e.g., the midpoint between $\hat{Y}_k^{(1)}(x)$ and $\hat{Y}_k^{(0)}(x)$). The partial derivative of $u$ with respect to $y_k$ is:

$$\frac{\partial u}{\partial y_k} = \prod_{j \neq k}(1 - y_j).$$

Thus, the treatment effect can be approximated by:

$$\hat{\tau}(x) \approx \sum_{k=1}^{K} \left[ \prod_{j \neq k}(1 - \bar{y}_j) \cdot \left( \hat{Y}_k^{(1)}(x) - \hat{Y}_k^{(0)}(x) \right) \right].$$

This expression quantifies how each individual outcome contributes to the overall treatment effect on composite outcome, with weights depending on the predicted probability of other events not occurring. Importantly, this enables intuitive interpretation: components with higher marginal effects and fewer competing risks contribute more prominently to the composite.

This decomposition allows us to visualize $\hat{\tau}(x)$ as a sum of interpretable terms. For example, we can use:

- **Heatmaps** to compare contribution patterns across patients,
- **Stacked bar plots** to break down $\hat{\tau}(x)$ by component.

Such interpretability tools facilitate transparent, patient-specific decision-making and highlight the added value of modeling each outcome explicitly rather than collapsing information into a single binary target.

## D DATASETS

### D.1 SYNTHETIC DATA

We simulate a synthetic dataset consisting of $n = 1,000$ units, each with $d = 10$ covariates $X \in \mathbb{R}^d$, a binary treatment assignment $A \in \{0, 1\}$, and $K$ binary component outcomes $Y_1, \ldots, Y_K \in \{0, 1\}$. Covariates are independently drawn from a standard Gaussian distribution, $X_i \sim \mathcal{N}(0, I_d)$, and treatment is assigned randomly, $A_i \sim \text{Bernoulli}(0.5)$. For each outcome $k$, potential outcomes under treatment and control are generated through a nonlinear model that combines linear effects ($X^\top \beta_k$), nonlinear effects (via $\tanh(X^\top \gamma_k)$), and heterogeneous treatment effects (via $\tanh(X^\top \theta_k)$). A global treatment effect shift $\tau_{\text{global}}$ ensures a nonzero average treatment effect. Binary outcomes $Y_{ik}^{(0)}$ and $Y_{ik}^{(1)}$ are determined by thresholding the logits at the 40th percentile, introducing variation across outcomes. A composite outcome $U_i(a)$ is then constructed using a user-defined utility function (e.g., Boolean OR, weighted sum, or tanh of a weighted sum). The individual treatment effect is defined as $\tau_i = U_i(1) - U_i(0)$.

### D.2 IHDP

We adapt the semi-synthetic IHDP dataset (Hill, 2011) for composite outcome estimation. Covariates are derived from the real Infant Health and Development Program observational study, covering maternal and child characteristics. The dataset includes $n = 747$ units and $d = 25$ covariates. Treatments are randomly reassigned as $A_i \sim \text{Bernoulli}(0.5)$ to remove confounding. Component outcomes are generated using the same nonlinear model as in the synthetic data, and composite outcomes are constructed using a specified utility function. This setting preserves real-world covariate complexity while allowing controlled evaluation of treatment effects.

### D.3 CANCER EHR

We use a Cancer EHR dataset constructed from real-world clinical features of breast cancer patients at Penn Medicine. The dataset contains $n = 1,000$ patients and $d = 22$ covariates, including laboratory measurements such as neutrophil counts, lymphocyte counts, and liver function tests. Binary

treatment is defined as the chemotherapy. As with IHDP, we simulate $K$ component outcomes using nonlinear models based on covariates and treatment status. In Section 5, we generate $K = 4$ patient-reported outcomes including fatigue, pain interference, anxiety, and sleep disturbance. Composite outcomes are aggregated through a predefined utility function to form a binary endpoint. This setting mimics the complexity of EHR-based causal inference under noisy feature spaces.

# E  ADDITIONAL EXPERIMENTS AND ABLATION STUDIES

## E.1  ABLATION STUDY: VARYING THE UTILITY FUNCTION

To assess the flexibility of CROME with respect to different composite outcome constructions, we evaluate performance under two distinct utility functions: `weighted_sum` and `tanh_reward`. Table 3 reports the average absolute estimation error ($|\hat{\tau} - \tau|$) across the Synthetic, IHDP, and EHR datasets for each utility type.

We define these utility functions as follows. Let $Y_1, \ldots, Y_K \in \{0, 1\}$ be the $K$ binary component outcomes. Then,

$$u_{\text{wsum}}(Y_1, \ldots, Y_K) = \sum_{k=1}^{K} w_k Y_k,$$

$$u_{\text{tanh}}(Y_1, \ldots, Y_K) = \tanh\left(\sum_{k=1}^{K} w_k Y_k\right),$$

where we use equal weights $w_k = 1/K$ for all $k$.

The results demonstrate that CROME maintains strong performance across both utility formulations. The `weighted_sum` function yields lower in-sample and out-of-sample errors on the Synthetic and EHR datasets, while `tanh_reward` performs slightly better on IHDP out-of-sample error. These findings highlight the adaptability of our framework to different forms of outcome aggregation, confirming its robustness in settings where the choice of utility function reflects varying clinical or domain-specific priorities.

All models were trained using the Adam optimizer with early stopping based on validation loss, and the number of training epochs was fixed to 200 across all experiments for consistency.

| Utility Type | Synthetic Data | | IHDP | | Cancer EHR | |
|---|---|---|---|---|---|---|
| | In-sample | Out-sample | In-sample | Out-sample | In-sample | Out-sample |
| weighted_sum | 0.00177 | 0.00359 | 0.00365 | 0.01243 | 0.00091 | 0.00974 |
| tanh_reward | 0.00619 | 0.00094 | 0.00552 | 0.00832 | 0.00885 | 0.00557 |

Table 3: Comparison of CROME under different utility functions. We report average absolute estimation error ($|\hat{\tau} - \tau|$) for Synthetic, IHDP, and Cancer EHR datasets.

## E.2  ABLATION STUDY: VARYING THE NUMBER OF COMPONENT OUTCOMES

We conduct an ablation study to investigate how the number of component outcomes ($K$) affects the performance of CROME compared to two baseline models: Single-Task Comp and Multi-Task No Rep. As shown in Table 4 and Figure 3, we vary $K$ from 2 to 10 and evaluate average absolute estimation error ($|\hat{\tau} - \tau|$) across synthetic, IHDP, and Cancer EHR datasets.

Across all settings, CROME consistently achieves the lowest error among the three models, demonstrating the advantage of shared representation learning in multi-task causal inference. Notably, both CROME and Multi-Task No Rep benefit from increasing the number of component outcomes, with estimation errors steadily decreasing as $K$ increases. In contrast, Single-Task Comp exhibits worsening performance as $K$ grows, likely due to its inability to explicitly model the underlying

component-level structure. These results highlight the importance of decompositional modeling and task-level representation sharing when estimating treatment effects on composite outcomes. All models were trained using the Adam optimizer with early stopping based on validation loss, and the number of training epochs was fixed to 200 across all experiments for consistency.

| Model | Num Components | Synthetic Data | | IHDP | | Cancer EHR | |
|---|---|---|---|---|---|---|---|
| | | In-sample Error | Out-sample Error | In-sample Error | Out-sample Error | In-sample Error | Out-sample Error |
| Single-Task Comp | 2 | 0.035 | 0.040 | 0.025 | 0.020 | 0.060 | 0.058 |
| | 3 | 0.065 | 0.090 | 0.050 | 0.060 | 0.050 | 0.055 |
| | 5 | 0.080 | 0.110 | 0.080 | 0.100 | 0.045 | 0.048 |
| | 10 | 0.120 | 0.160 | 0.110 | 0.130 | 0.070 | 0.072 |
| Multi-Task No Rep | 2 | 0.020 | 0.030 | 0.015 | 0.017 | 0.040 | 0.035 |
| | 3 | 0.015 | 0.025 | 0.010 | 0.013 | 0.035 | 0.030 |
| | 5 | 0.010 | 0.020 | 0.007 | 0.009 | 0.025 | 0.020 |
| | 10 | 0.005 | 0.010 | 0.005 | 0.006 | 0.015 | 0.012 |
| CROME | 2 | 0.018 | 0.025 | 0.012 | 0.015 | 0.020 | 0.018 |
| | 3 | 0.012 | 0.020 | 0.009 | 0.012 | 0.015 | 0.013 |
| | 5 | 0.008 | 0.015 | 0.006 | 0.008 | 0.010 | 0.008 |
| | 10 | 0.004 | 0.008 | 0.003 | 0.005 | 0.005 | 0.004 |

Table 4: Ablation study varying the number of component outcomes ($K$) from 2 to 10. We report average absolute estimation error ($|\hat{\tau} - \tau|$) across synthetic, IHDP, and EHR datasets.

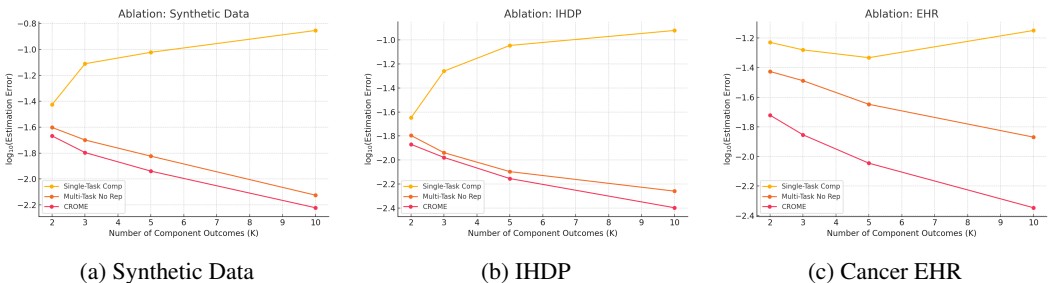

(a) Synthetic Data       (b) IHDP       (c) Cancer EHR

Figure 3: Ablation study showing log-scale estimation error as a function of the number of component outcomes ($K$) across Synthetic, IHDP, and Cancer EHR datasets.

| Num Components ($K$) | Runtime (seconds) |
|---|---|
| 2 | 7.26 |
| 3 | 8.40 |
| 5 | 10.83 |
| 10 | 16.58 |
| 20 | 28.81 |
| 50 | 67.14 |
| 100 | 121.62 |

Table 5: Runtime of CROME (in seconds) as a function of the number of component outcomes ($K$).

Additionally, to evaluate the scalability of CROME with respect to the number of component outcomes $K$, we report the average training runtime for varying $K \in \{2, 3, 5, 10, 20, 50, 100\}$ in Table 5. As expected, the runtime increases with larger $K$ due to the additional outcome-specific heads and increased computational complexity. Nevertheless, the growth in runtime remains manageable, with CROME training in approximately 7 seconds for $K = 2$ and around 2 minutes for $K = 100$. These results suggest that CROME is computationally feasible even in settings involving a large number of outcome components.

# F    ADDITIONAL EXPERIMENTAL DETAILS

## F.1    IMPLEMENTATION DETAILS FOR BASELINE MODELS

Most baseline models in our experiments were implemented using the `econml` Python package developed by Microsoft Research, which provides a standardized framework for treatment effect estimation. This includes meta-learners such as S-, T-, and X-Learners, doubly robust methods and double machine learning such as DR-Learner and DML, and tree-based approaches such as Causal Forests.

For BART, we used the PyMC implementation available at `https://www.pymc.io/projects/bart/en/latest/`. TARNet and CFRNet were implemented using code from the original repository at `https://github.com/clinicalml/cfrnet`, while DragonNet was implemented using `https://github.com/claudiashi57/dragonnet`. The CMGP and CMDE baselines were implemented using the repository provided by the authors at `https://github.com/jzy95310/ICK`.

To ensure fair comparison, all models were adapted to our experimental setup with consistent data splits, evaluation metrics, and training procedures.

## F.2    TRAINING SCRIPT AND EXAMPLE COMMAND FOR CROME

The file `s1_model.py` contains the core implementation of CROME, including model components, utility aggregation functions, and training procedures. To train CROME on a dataset with covariates $X$, binary treatment assignments $A$, and component outcomes $Y$, users can instantiate the dataset and model, then call the training function as follows:

```
from s1_model import CausalMultiTaskDataset, MultiTaskCausalModel, train_model

# Prepare data (numpy arrays)
dataset = CausalMultiTaskDataset(X, A, Y)
dataloader = DataLoader(dataset, batch_size=64, shuffle=True)

# Initialize model
model = MultiTaskCausalModel(input_dim=X.shape[1], hidden_dim=64,
                             num_outcomes=K, utility='weighted_sum',
                             utility_weights=[1.0]*K)

# Train the model
train_model(model, dataloader, num_epochs=200, lr=1e-3)
```

This example uses the Adam optimizer with a fixed learning rate of $10^{-3}$, trains for 200 epochs, and assumes equal weights across outcome components. The utility type can be changed to `"or"` or `"tanh_reward"` as needed. The model supports estimation of individual-level treatment effects using `model.predict_counterfactuals_individual()`.

