# OpenReview forum: "CROME: Covariate-Balanced Causal Representation Learning for Composite Outcomes via Multi-Task Estimation in Electronic Health Records"
_ICLR.cc/2026/Conference — ICLR 2026 Conference Withdrawn Submission_

### Official Review · Reviewer_ojfD · 2025-10-15

**Soundness:** 2
**Presentation:** 2
**Contribution:** 1
**Rating:** 2
**Confidence:** 3

**Summary:**

The work introduces CROME, a causal representation learning framework that estimates treatment effects on composite outcomes via multi-task learning with covariate balancing. Different to single-task or unshared multi-task models, it jointly learns shared representations across outcome components. In experiments on synthetic, IHDP, and cancer EHR datasets, CROME achieves lower estimation errors. Further, it enables decomposition of treatment effects into outcome-specific contributions.

**Strengths:**

- CROME leverages shared representations and covariate balancing to achieves lower estimation errors than traditional single-task or unshared multi-task models.
- The method seems to work in practice.
- CROME enables decomposition of treatment effects into outcome-level contributions.

**Weaknesses:**

- **Incremental novelty**:
The core ingredients (causal representation learning with covariate balancing + multi-task learning for causal effects) are well-studied separately. The paper frames their integration for composite outcomes as “underexplored”. To me, this reads as a synthesis rather than something really novel.

- **Theory builds on existing bounds**:
From my understanding, the generalization guarantees reuse established multi-task bounds (Maurer et al., 2016). Hence, there is limited theoretical novelty beyond adapting known results to the composite-outcome setting.

- **Utility-functions**:
Results emphasize the Boolean-OR utility. However, there are limited ablations to weighted-sum/tanh. Broader utilities common in practice (specifically, non-separable utilities) are not sufficiently evaluated. Unfortunately, this strongly narrows the applicability.
To be more specific, theory and experiments focus on Lipschitz-separable utilities like Boolean OR or weighted sums.
However, in many real clinical settings, the composite outcomes depend on complex interactions among outcomes. To me, it is not clear whether the shared representation and generalization guarantees would still hold or how easily the method could adapt to these non-separable utilities.

- **Overlength**:
Including discussion, the paper goes over 9 pages.

- **Writing and presentation**:
The paper would significantly benefit from improving writing and presentation.

- **Interpretability claims**:
The strongly promoted interpretability claims in the paper (Sec. 4.3) are very basic.

____

Andreas Maurer, Massimiliano Pontil, and Bernardino Romera-Paredes. The benefit of multitask
representation learning. Journal of Machine Learning Research, 17(81):1–32, 2016.

**Questions:**

**Novelty Clarification:** How does CROME fundamentally differ from prior causal multi-task frameworks such as CMGP or CMDE. To me, it seems they are simply applied to composite outcomes?

**Theoretical Contribution:** The generalization bound builds on Maurer et al. (2016); what new theoretical insights or assumptions are introduced specifically for causal inference on composite outcomes?

**Utility Function Flexibility:** How well does CROME handle non-separable (such as non-additive) or continuous-time composite utilities (such  as survival-based composites)?

____

Andreas Maurer, Massimiliano Pontil, and Bernardino Romera-Paredes. The benefit of multitask
representation learning. Journal of Machine Learning Research, 17(81):1–32, 2016.

---

### Official Review · Reviewer_oTSX · 2025-10-16

**Soundness:** 1
**Presentation:** 1
**Contribution:** 1
**Rating:** 2
**Confidence:** 4

**Summary:**

In this paper, authors have proposed a novel framework for handling composite (ie multiple simultaneous) outcomes for individualised treatment effect (ITE) estimation under binary treatment setting. The idea is to use multitasking where we have shared layers to learn a common representation and task/outcome specific layers for each of the component of the composite outcome. They learn balanced representations using the approach as discussed in the pioneering work of TARNet/CFRNet paper, where instead of one outcome, we now have multiple outcomes. The key idea of the framework, which makes it different from the TARNet paper, is that it uses a user-defined utility function, after training, to combine the multiple simultaneous outcomes of the composite outcome into a single outcome, and thus has a single number/treatment effect rather than multiple outcomes. The authors claim that this would enable interpretability, as we can see how different outcomes are contributing to the composite outcome, as compared to training a model on the composite outcome directly. They provide theoretical results to prove that the proposed approach has better generalisation error as compared with training a single composite task or training each outcome independently. Then, they provide sufficient empirical analysis to justify these claims.

**Strengths:**

1. The paper proposes a novel framework for ITE estimation under a binary treatment setting which can handle composite outcomes. The basic idea is to use a multitasking like structure for training and afterwards a user defined utility function is used to combine different outcomes. Thus, instead of several outcomes, now we would get a single number and can also clearly see the contribution of different outcomes to the composite outcome. This is novel in the sense that, as the authors claim and to the best of my knowledge, this is an interesting way (but not necessarily sufficiently novel, as discussed below) for dealing with composite outcomes.

2. The paper provides theoretical bounds which prove that the proposed approach results in tighter error bounds as compared to training all outcomes independently, as well as training a composite task directly by first combining the outcomes using a utility function (not verified the theorems).

3. They provide empirical analysis on synthetic and two semi-synthetic (IHDP and cancer EHR) datasets. This analysis does support their claims of interpretability and better results as compared to baselines. The authors think that the interpretability claims are clinically very useful.

**Weaknesses:**

My major concerns are related to the novelty of the work and the way it is presented against the existing work, as discussed below.

1. We already have papers using multitasking for ITE, as cited by the authors, and the pioneering work TARNet/CFRNet, which employed covariate balancing as used by the authors. To my understanding, this paper extended the CFRNet paper from a single task/outcome to multiple outcomes/tasks and then, once such a model was trained, they combined those multiple outcomes using a user defined utility, which helps get a single composite treatment effect as compared to getting treatment effects on multiple outcomes. I acknowledge that this was the novel bit, but this does not seem sufficient to me. Moreover, the utility function comes into play once the training is completed, which is not significant for the venue.

2. The paper presents itself as existing research applies a user defined utility function before training to get a single composite outcome and then ITE estimator is trained on that single outcome. Then, their results compare against this single outcome as well as training each outcome independently. My first question is, why were people doing so? Did you fix something that enables you to train multiple outcomes together? Multitasking ideas were there, why were people not using them? To my understanding, you did not fix anything that enables you to train multiple outcomes simultaneously. So, the way paper presents itself to fixing/solving the composite outcome problem does not make sense to me. Second, it is well known in ML literature that multitaksing improves results against training each task independently under some conditions. So, empirical and theoretical results proving this point also do not make sense to me. That means using multitasking for multiple outcomes is most likely meant to give better results than training each outcome independently.

3. The paper presents the idea of using a utility function ambiguously. From the paper's discussion, it appears that utility function is part of the end-to-end training and that would have been something very interesting and stronger point, but by looking at the loss functions, it is clear that the utility function was not part of the training for the proposed model.

4. It appears that this work discusses a new framework to combine composite outcomes. However, in the related work, the authors have used a very vague terminology, saying that it is underexplored, without discussing the existing work to deal with it. This makes the work misplaced wrt literature, as it is not clear where the existing methods fail. So, either explicitly claim the novelty or compare the proposed method against the existing methods in a table to show the novelty of the work.

5. The introduction does not cite/discuss any of the existing works in context to discuss the limitations of the existing works to motivate the idea.

6. Other points needing clarity:
- some clarity is needed around composite outcomes, components, prediction task, single-task, task vs outcome etc. Use of MTL was also confusing as MTL generally means having shared layers. Reader gets a clear picture about these related terms quite late in the paper, which affects the readability.

- There is a clash of notations: Y_k(a) is used to define an indicator for a component k as well as potential outcomes on page 3.

- How practical is it to put Kx2 heads, especially for a large number of outcomes (e.g. 100 as used by you)? Moreover, this would not work with continuous treatments?

- Use of the Taylor expression equation on 352, i.e. on the utility function, is also not very clear.

**Questions:**

1. Why is the loss function part of the preliminaries rather than part of the model?
2. You mentioned that you extended the baselines to multitasking setting for comparison. How does your method differ from CFRNet when you extend that to multitasking. During training they should look the same because you extended the idea from that paper to multitasking with a utility funciont used after training. Did I miss anything?
3. How does the outcome look after combining for a single task using a utility function. Since the utility function was using indicator variables so is the outcome still a regression problem?

---

### Official Review · Reviewer_QB1r · 2025-10-20

**Soundness:** 3
**Presentation:** 3
**Contribution:** 3
**Rating:** 6
**Confidence:** 3

**Summary:**

The paper proposes a multi-task learning strategy for estimating treatment effects with composite outcomes. The main contribution is that it provides a theoretical view why Multi-task learning and shared representation can perform better on the composite outcomes with a proved error bound for the estimation.

**Strengths:**

- The paper provides an insight into why MTL can help improve the treatment effects estimation on composite outcome, and provide both theoretical and observational evidence to support their claims.
- Another theoretical contribution of the paper is that it shows evidence that with sharing representation for MTL in this scenario, the model can perform better.
- The application scenario is somewhat novel. Composite outcomes are rarely studied in treatment effects estimation, and the scenario is common in health care domain.
- One of the contributions as claimed by the paper is that the multi-task learning framework allows to interpret the model by decomposing the estimation using talor expansion to check how each component of the outcome contribute to the composite outcome. It is an interesting way to enhance model's interpretability from theoretical point of view (But I have concerns about the application in real world problem, please see weakness 3 bellow).

**Weaknesses:**

- Although the paper claims that the proposed framework can adapt to any Lipschiz continuous outcome $u(.)$, the 3 observational studies only talks about the Boolean OR utility function. To support your claims, I suggest the author experiment more on some other utility function cases on either synthetic data or IHDP.
- Although the idea of the decompostion of estimation into outcomes' effects. In the context of causal inference, we care about pre-intervention's hetergeneity for interpreting model and thus can help decision making.
- The theory as provided the paper is based on binary treatment and composite binary outcomes setting. But numerical measurements and oridinal score values are very common in the healthcare applications. There is no such discussion about how will the theory gerneralized to such settings.

**Questions:**

- As to one of the advantages as claimed by the paper, one can decompose the esimation of the treatment effects to investigate how each component of the outcomes contributes to the final effects. Figure 2(b) shows that the decomposition can lead to variours effects of different components of the outcome to the treatment effects, however, in causal inference, outcome should always serve as the observation after intervention. My question is when treating patients in future, with no observation of the outcome, how can your model be used to guide clinical decisions among individuals? For interpretation purpose in this field, we still care about how each individual's characteristics contribute to the effects.
- IHDP is a well known dataset for benchmarking TE, but in your new scenario, the constructed new outcome lable is confusion. line 376-377: How do you generate binary outcomes using covariates? Need more details here to make sure the experiment set up is fair.

---

### Official Review · Reviewer_QYUY · 2025-11-01

**Soundness:** 2
**Presentation:** 1
**Contribution:** 2
**Rating:** 2
**Confidence:** 4

**Summary:**

The paper introduces CROME, a causal representation framework for composite outcomes. It learns a shared encoder with outcome- and treatment-specific heads, adds a covariate-balance regularizer in the latent space, predicts component-level potential outcomes, and aggregates them via a user-specified utility to obtain composite CATE/ATE. Experiments are conducted on synthetic data, semi-synthetic data, and a cancer EHR setting with simulated PROs, where CROME outperforms a wide set of baselines.

**Strengths:**

- Composite endpoints are common in healthcare; modeling components then aggregating is well-motivated and improves interpretability.

- Experiments are conducted on three different datasets against multiple baselines.

**Weaknesses:**

- The core proposed components (MTL with shared encoder, IPM/MMD balancing, component prediction with utility aggregation) are individually well-studied and established; the combination is incremental. The paper needs to clearly demonstrate its technical contribution against CFRNet/DragonNet (representation balancing), multi-label causal and multi-task causal GP, etc.

- The method assumes strong ignorability/positivity but lacks sensitivity analyses to unobserved confounding and overlap diagnostics. Claims of “principled” would be stronger with robustness checks.

- The EHR experiment uses simulated PROs and a Boolean-OR composite; no real composite outcome is evaluated, and utility misspecification is not tested. This limits external validity.

- It’s unclear whether all baselines received matched hyperparameter tuning process for a fair comparison. Some reported errors appear unusually high for certain baselines (e.g., CFR on IHDP).

- The overall presentation of the paper is weak and requires clarification. For example, Figure 2 is of low visual quality, there are numerous typos (e.g., “strong ignobility”), and several methodological details, such as the choice of kernel for MMD, the parameterization of the heads, and the specification of head loss functions, are insufficiently explained.

**Questions:**

- Provide overlap diagnostics (e.g., propensity score histograms in latent space), and ablations under induced hidden confounding.

- Choice of IPM: Compare MMD vs. Wasserstein/IPM variants.

- The simulation setup is overly simplistic and fails to reflect real-world complexity. It assumes near-additive effects, weak interactions, and ideal overlap, which may artificially favor CROME. More realistic scenarios with correlated components, nonlinear confounding, imperfect overlap, and noise heterogeneity would better validate the method’s robustness.

---

### Note · Authors · 2025-11-19

I have read and agree with the venue's withdrawal policy on behalf of myself and my co-authors.